# Towards Neural Architecture Search through Hierarchical Generative Modeling

## Abstract

Neural Architecture Search (NAS) is gaining popularity in automating designing deep neural networks for various tasks. A typical NAS pipeline begins with a manually designed search space which is methodically explored during the process, aiding the discovery of high-performance models. Although NAS has shown impressive results in many cases, the strong performance remains largely dependent on, among other things, the prior knowledge about good designs which is implicitly incorporated into the process by carefully designing search spaces. In general, this dependency is undesired, as it limits the applicability of NAS to less-studied tasks and/or results in an explosion of the cost needed to obtain strong results. In this work, our aim is to address this limitation by leaning on the recent advances in generative modelling – we propose a method that can navigate an extremely large, general-purpose search space efficiently, by training a two-level hierarchy of generative models. The first level focuses on micro-cell design and leverages Conditional Continuous Normalizing Flow (CCNF) and the subsequent level uses a transformer-based sequence generator to produce macro architectures for a given task and architectural constraints. To make the process computationally feasible, we perform task-agnostic pretraining of the generative models using a metric space of graphs and their zero-cost (ZC) similarity. We evaluate our method on typical tasks, including CIFAR-10, CIFAR-100 and ImageNet models, where we show state-of-the-art performance compared to other low-cost NAS approaches.

## 1 Introduction

Designing more accurate and efficient deep neural networks (DNNs) has been one of the primary areas of focus in machine learning research in recent years. However, developing a novel architecture is often accompanied by burdensome trial-and-error in an attempt to find an *optimal* DNN configuration. More recently, this process has been aided by a broad family of algorithms designed to automate optimizing a machine learning (ML) system. Among those, Neural Architecture Search (NAS) (Zoph & Le, 2017) emerged as a subfield of ML focused specifically on finding the best working DNN from a pre-defined discrete set of candidates (often referred to as a *search space*). While NAS approaches have proven effective in many cases, often noticeably improving upon manual efforts, their usability remains limited by several challenges (White et al., 2023), from search efficiency and robustness to transferability across different datasets and/or tasks. In this paper, we focus on advancing NAS by tackling one of the challenges that remain somewhat overlooked, possibly due to its particular complexity – the challenge of being limited by the initial NAS search space, which is often rigid and hand-crafted. In practice, although carefully designed search spaces can improve NAS performance (Radosavovic et al., 2020), e.g. decreasing search time or improving the chance of finding good-performing architectures, it limits the applicability of NAS approaches to those tasks where strong/sensible search spaces are available. More importantly, by definition, the performance of any NAS approach is strictly associated with the quality of its search space – even the best searching algorithm can only produce results as good as the best candidate model from within the search space.

The existence of this inherent relationship between the statistical properties of a search space and the cost-performance trade-off in NAS is well-known. It has motivated researchers to explore the problem of designing better search spaces, resulting in several methods attempting to automate this process (Zhou et al., 2021; Hu et al., 2021; 2020). Generally speaking, automated search space design methods can be seen as mappings from one search space to another, $S \rightarrow A$, where typically $A \subset S$ and $|A| \ll |S|$. In other words, these methods focus on identifying a subset ($A$) of a larger search space ($S$) that exhibits desired statistical properties, which is then used by a chosen NAS algorithm to find individual models. Compared to performing NAS directly on $S$, they are more efficient due to the relaxed objective in the first stage, effectively resulting in hybrid systems combining coarse and fine-grained searching. However, these approaches still need to carefully design the initial search space $S$ (e.g., as a pre-defined super network in Zhou et al. (2021)), limiting the final NAS

performance in a way similar to classical methods. On the other hand, although we can make $S$ much larger and still search efficiently via automated search space design, this is not necessarily scalable to the levels that would allow us to avoid design bias in general. For example, existing approaches that focus on pruning or evolving a large initial $S$ often rely on repeating a variation of the NAS process multiple times to provide feedback for updating $S$. This is also subject to bias, e.g. when assessing the fitness of different subspaces, and will almost certainly result in non-negligible additional cost (Zhou et al., 2021; Ci et al., 2021).

In contrast, our goal is to scale the initial search space $S$ to sizes unattainable by previous works, in an attempt to provide a more general-purpose mechanism with less dependence on the initial design of $S$. Simply put, we want to include as many models in $S$ as possible, without worrying too much about the resulting computational cost or human effort of doing so. In order to navigate the resulting large space efficiently and in a context-dependant way, we are inspired by the recent advances in generative ML (Grathwohl et al., 2019) which shows impressive results in modelling highly-dimensional conditional distributions of various modalities such as images, text or speech. To extend this high-level idea to distributions of neural networks, we propose a hierarchical approach which breaks down the generation into several steps: *1)* first, we learn a reversible, continuous latent space of micro/cell designs in which "similar" designs cluster together – this is achieved by training a Graph Variational Autoencoder (G-VAE) regularized by the zero-cost (ZC) similarity of different graphs; *2)* then, we introduce a Conditional Continuous Normalizing Flow (CCNF) model as a way of finding "synonymous designs" in the G-VAE space efficiently; *3)* finally, a decoder-only sequence generator (SG) is trained to design macro architectures – for a user-defined conditioning it outputs a sequence of cells to form a full model, with architectural details of individual cells being abstracted away through the notion of a "cell type" obtained from the previous steps. The full process is schematically depicted in Fig. 1 with Sec. 3 discussing each step in detail. In summary, our work contributes in the following ways:

- We extend existing ZC proxies into a multidimensional vector form, studying their clustering behaviour on network graphs and their effectiveness.
- We employ semi-supervised learning for a G-VAE to achieve a reversible encoding of graphs into a latent space, preserving the property of the ZC space and we further perform clustering. This approach further enables us to quantize the graph design space into multiple "synonyms" sets without compromising graph performance and costs.
- We use a CCNF model to efficiently navigate the latent space of the G-VAE, which results in efficiently generated candidate cell (graph) architectures under specific conditions.
- We construct a decoder-only transformer to integrate with graph generation, forming a hierarchical network architecture generation mechanism that supports user-defined conditioning.

## 2 RELATED WORK

**Automated NAS Search Space Design.** Optimizing NAS search spaces in an automated manner has attracted extensive interest recently, e.g. by progressively constraining the degree of freedom of the network design space (Radosavovic et al., 2020), interchangeably shrinking and expanding the initial search space (Ci et al., 2021), or employing evolutionary algorithms to evolve the initial space to an optimal subspace Zhou et al. (2021). Schrodi et al. (2023) propose a unifying framework to design hierarchical NAS search spaces with context-free grammars and use Bayesian optimization to search efficiently. Despite their differences in how to find the optimized subspaces, most of the existing approaches still need to iteratively evaluate the fitness of subspaces, i.e., assessing the performance of models within them, which often results in significant cost. A large body of work has been developed to speed up model performance evaluation in general NAS context (Pham et al., 2018; Dudziak et al., 2020; Abdelfattah et al., 2021; Mellor et al., 2021; Xiang et al., 2023a), e.g., zero-cost proxies (Abdelfattah et al., 2021; Mellor et al., 2021; Xiang et al., 2023a) have been shown to correlate well with final trained accuracies, with negligible cost to compute.

**Flow-based Models.** Compared to other types of generative models (Creswell et al., 2018; Kingma & Welling, 2013), flow-based models are more efficient to sample, while being more stable to train and free of mode collapse and convergence issues, showcasing impressive performance in various generation tasks (Abdal et al., 2021; Yang et al., 2019; Li et al., 2022). Existing works often consider the continuous normalizing flows (CNF) (Grathwohl et al., 2019) based on neural ODEs (Chen et al., 2018), which are able to impose conditions (e.g., attributes) during the sampling process, and thus control certain properties of the generated output, e.g., generating attribute-semantic edits (Abdal et al., 2021) or super-resolution (Lugmayr et al., 2020) of images. Our work is based on the existing work on flow-based models, but to the best of our knowledge, we are the first to show that it is feasible to employ CCNFs to generate optimized neural architectures, balancing performance and efficiency.

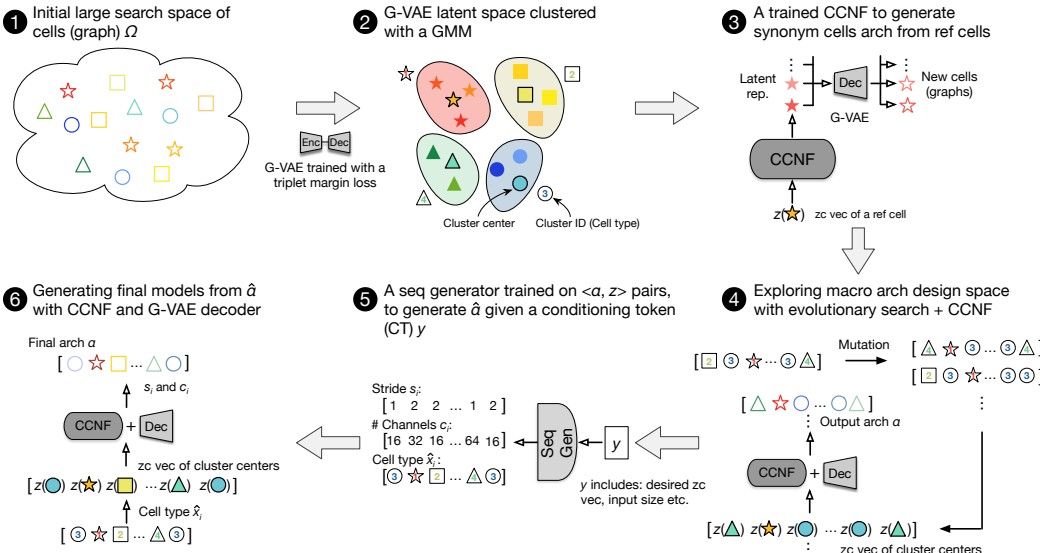

Figure 1: Overview of the proposed approach. ❶ We start from a large micro search space $\Omega$, and ❷ train a G-VAE with triplet margin loss and use a GMM to partition the latent space into clusters. ❸ A CCNF model is then trained to generate synonym cells given a reference $r$. ❹ We explore the high-level search space $\hat{S}$ with ageing evolution and the trained CNF, and ❺ the set of explored architectures $\{\alpha\}$ is used to train a SG, which is able to generate the desired $\hat{\alpha}$ given a condition token (CT) $y$. ❻ The final architecture is then generated from $\hat{\alpha}$ using the CCNF and G-VAE decoder.

**Neural Architecture Generation.** Neural architecture generation (NAG) has recently become a popular topic in the NAS community (Lee et al., 2021; Yu et al., 2023; An et al., 2023). Compared to conventional NAS, NAG methods often resort to some forms of generative models, such as diffusion models (An et al., 2023) or Generative Pre-Trained (GPT) models (Yu et al., 2023), aiming to generate candidate architectures with desired properties. Our work bears some resemblance to these methods, but are fundamentally different: *1)* instead of direct search/generation, we consider a hierarchical paradigm to efficiently explore the tremendously large initial search space; *2)* we consider a synonymous micro design approach leveraging the clustering properties of architectures, to effectively discover high-performing cells from given references; and *3)* we use a vector form of zero-cost proxies throughout the micro and macro design, achieving desired trade-off between cost to compute and transferability across different tasks.

## 3  METHOD

This section outlines the technical details of our method and search space. We begin by discussing clustering of similar micro designs in an invertible fashion in Sec. 3.1, then follow with synonym generation using a CCNF model in Sec. 3.2. Finally, we discuss full model generation with a decoder-only transformer in Sec. 3.3, while the search space details can be found directly below.

**Micro search space.** Our micro design space consists of any directed graph with up to 6 operation nodes, 1 input node and 1 output node. Operation nodes can be assigned one of 28 operations from the literature (9 op. types with 3 kernel size each + skip connection), and the $n^{\text{th}}$ node can have up to $n$ inputs – one for each previous node; values of different inputs are added before executing a selected operation. Each cell design includes the number of channels and stride as hyperparameters – operations within a cell are not allowed to change dimensions of intermediate results (unless internal to an operation), and a relevant projection is performed automatically before executing a cell to adjust input channels and possibly reduce spatial dimensions. Overall, without accounting for isomorphism and cell hyperparameters, the micro design space spans approximately up to $10^{17}$ configurations. Formally, the micro space is a set of pairs $\Omega = \{(\mathbf{A}, \mathbf{F})\}$ where $\mathbf{A} \in \{0,1\}^{8\times8}$ is an adjacency matrix and $\mathbf{F} \in \{0,1\}^{8\times13}$ is a feature matrix encoding operation type and kernel size (1-hot over the first 10 and last 3 dimensions). Details about operations can be found in App. B.

**Macro search space.** The macro search space is then defined as a sequence of up to 20 different cells $x_i \in \Omega$, with each cell being additionally parametrized by the number of channels $c_i \in C = \{8, 16, ..., 1024\}$ and stride $s_i \in \{1, 2\}$, for $i = 1, 2, ..., 20$. If we assume different triples

$t_i = [x_i, c_i, s_i]$ to be unique designs, the overall search space is then a set of all sequences of any 20 such triples $S = (\Omega \times C \times \{1, 2\})^{20}$ and we can estimate the total number of points to be up to $(|\Omega| * 127 * 2)^{20} \approx 10^{390}$. We will use $\alpha(t_1, t_2, ..., t_{20}) \in S$ to refer to particular design points in the overall search space.

## 3.1 CLUSTERING MICRO DESIGNS

Training a macro model generator to find high-quality models directly in such a large search space is obviously challenging. In order to simplify this task, we want to abstract away details of individual designs by automatically identifying families of similar cells. Later, the sequence generator only has to decide a family for each cell and the task of finding high-performance designs within each selected family is delegated to a lower-level search in the hierarchy.

Formally, given a metric space of micro designs $(\Omega, d)$, where $d(a, b)$ is a distance metric between designs $a$ and $b$ proportional to their relative quality, our goal is twofold: *1)* to partition $\Omega$ into a finite number of $K$ clusters $\hat{\Omega}_{k=1,...,K}$ based on the metric $d$; *2)* to be able to sample new designs from each cluster easily.

The first important choice here is the metric $d$ - naively relying on accuracy is undesired for two reasons: *1)* computing accuracy for a large number of models can be extremely expensive, a common challenge in NAS, but also *2)* it is known that different models can exhibit very different performance on different downstream tasks (Duan et al., 2021; Tu et al., 2022) – since we want to avoid bias towards any particular downstream task, we need something else.[1] To address this, we propose to use a generalized form of ZC proxies instead, by computing a vector of $P$ proxies for a semi-fixed point from $S$. Specifically, for a micro design $x \in \Omega$, we define its ZC vector $z : \Omega \to \mathbb{R}^P$ as:

$$z(x) = \left[ z_{\alpha_x}^{(1)}, z_{\alpha_x}^{(2)}, ..., z_{\alpha_x}^{(P)} \right], \quad \text{where:} \quad \alpha_x = \alpha(t_1, t_2, ..., t_{17}) \quad \text{s.t.} \tag{1}$$

$$t_i = [x, c_i, s_i], \quad c_i = 16 * 2^{\lfloor \frac{i}{6} \rfloor}, \quad s_i = \begin{cases} 2 & \text{if } i \in \{6, 12\} \\ 1 & \text{otherwise} \end{cases} \tag{2}$$

and $z_{\alpha_x}^{(i)}$ refers to computing the $i^{\text{th}}$ zero-cost proxy for the model $\alpha_x$ obtained by stacking the same cell design into the above predefined macro structure. Note that the ZC vector, while related to the original ZC metrics, serves a fundamentally different purpose – instead of strictly ordering models, which is a challenging task, we only use it to cluster designs that are likely to train to similar performance. Although the difference might seem nuanced, it is crucial for our method. Consider a simple proxy of the number of parameters. For example, sometimes very large models are desired if working with large datasets, and sometimes small ones might be a better option. While there is no universal answer to choosing the right model size, we can reasonably expect if one small model works well, other small models should not be fundamentally wrong choices, and vice versa. This reasoning is the foundation for our design clustering. Given $z$, the metric $d$ is then defined as $d(a, b) = L1(z(a), z(b))$. We will refer to this particular form as ZC-similarity.[2] In our experiments, we use a vector of 4 common proxies: *1)* NASWOT (Mellor et al., 2021), *2)* SNIP-SSNR (Xiang et al., 2023b), *3)* number of parameters (Ning et al., 2021), *4)* FLOPS.

Following the choice of the metric $d$, we employ a Gaussian Mixture Model (GMM) to identify different clusters in the ZC space. Specifically, we fit a mixture of multivariate normal distributions, parameterized with factors $\pi_i$ and their means and variances $\mu_i, \sigma_i$, to the observed values of $z$:

$$p(z) = \sum_{i=1}^{K} \pi_i \mathcal{N}(z|\mu_i, \sigma_i), \tag{3}$$

using the expectation-maximization algorithm. After that, models are assigned to clusters based on which component of the mixture they are most likely to belong to – this means one cluster is modelled by one component of the GMM with $\mu_i$ as its center. The number of clusters is decided by plateauing Bayesian and Akaike information criterions.

The above provides us with a way of identifying families of cells, as shown in Fig. 2(a), but navigating the space $\Omega$ is still hard since we do not have a way of quickly providing designs that would belong

---

[1]Conceptually, we could also consider a vector of accuracies on a diverse set of uncorrelated tasks, but that becomes even more expensive.

[2]Strictly speaking, ZC-similarity should be understood as the opposite of the $L1$ distance since similar models should be close, *i.e.*, $\uparrow$ similarity $=\downarrow L1$.

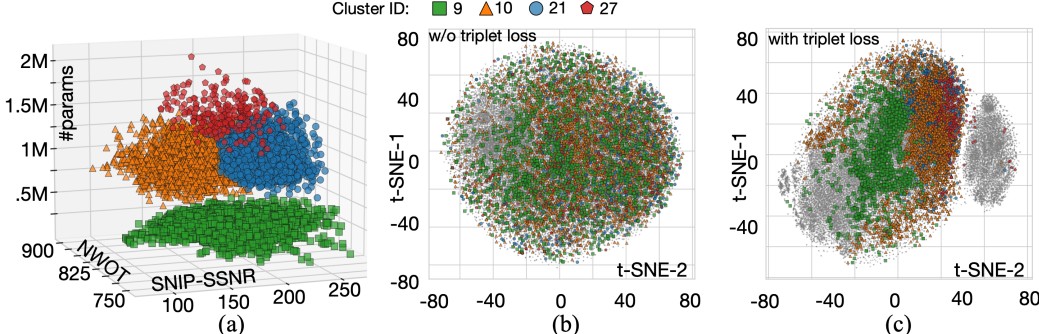

Figure 2: (a) Example of 4 clusters of micro designs identified with a GMM w.r.t. the ZC vectors $z$. (b) t-SNE visualization of the latent space obtained from a G-VAE trained naively vs. (c) using triplet loss regularization (the same 4 clusters highlighted). Unregularized space does not preserve ZC similarity of the models.

to a certain family. To address this, we further learn an invertible mapping to a continues latent space $f : \Omega \to \mathbb{R}^m$ in which $L1$ distance approximates $d$, *i.e.*, $(F, L1) \approx (\Omega, q)$, where $F = \{f(x)\}_{x \in \Omega}$ is the set of embedded designs in this latent space and $\forall_{a,b \in \Omega} L1(f(a), f(b)) \propto d(a, b)$. This is done by training a G-VAE model (Kipf & Welling, 2016) additionally regularized with a triplet margin loss (Balntas et al., 2016) and an auxiliary ZC predictor $q$ which tries to predict $z(x)$ from $f(x)$:

$$\mathcal{L}_{\text{VAE}} = - \overbrace{\mathbb{E}_{p(f|x)} \log p(\tilde{x}|f)}^{\text{reconstruction (BCE)}} + \overbrace{\text{KL}[p(f|x) \,\|\, p(f)]}^{\text{VAE regularization}} \tag{4}$$

$$+ \underbrace{\mathbb{E}_{x,a,b} \max \left\{ d(x, a)^2 - d(x, b)^2 + m, 0 \right\}}_{\text{triplet margin}} + \underbrace{\mathbb{E}_x (q(f(x)) - z(x))^2}_{\text{predictor (MSE)}}, \tag{5}$$

where $p(f|x)$ and $p(\tilde{x}|f)$ are distributions of encoded and reconstructed points, respectively; $m$ is a margin term and triples $x, a, b \in \Omega^3$ are chosen s.t. for an anchor $x$, $a$ and $b$ are positive and negative examples, respectively. In practice, to ensure the decoder of the G-VAE can correctly decode as many points in the latent space as possible, it is important to scale its training set as much as possible. However, computing $z(x)$ and related distances $d(a, b)$ might easily become a bottleneck; therefore, we decided not to compute the regularization term in Eq. 5 all the time and only limit it to a relatively small random subset of all training points. After the G-VAE is trained, we can approximate $f$ and $f^{-1}$ with its encoder and decoder, respectively.

Given the G-VAE and the GMM, we can efficiently realize all required functionality of identifying clusters, finding a cluster to which a model belongs and sampling from clusters. The resultant latent space and its clustering properties w.r.t. $z$ when trained naively vs. using our proposed regularization is illustrated in Fig. 2(b)-(c) – clearly, the regularized space preserves original clustering much better.

### 3.2 GENERATING SYNONYMOUS MICRO DESIGNS

At this point, we have identified $K$ different families of cells and have an easy way of sampling from $\Omega$ in a structured way thanks to our latent space. However, recall that we aim to be able to find high-performance designs in each family efficiently – this is still challenging since $K \ll |\Omega|$, meaning each cluster is still extremely big. To make the problem more approachable, for now, we will relax it to the problem of robustly finding synonyms of different designs – assuming we already know an example high-performance reference design $r$, we want $p(x|r)$ to be a distribution of designs with their accuracy centred around that of $r$. This problem is much simpler and should in fact be solvable by directly leveraging the properties of our latent space. In particular, $p(x|r)$ could be realized by sampling a neighborhood of $r$ in the latent space: $f^{-1}(f(r) + \epsilon), \epsilon \sim \mathcal{N}$. However, we observe that even in a simple case of CIFAR-10 classification as a downstream task, a naive sampling strategy like this leaves some room for improvement, which we attribute to the sheer amount of possible models and imperfections of the VAE latent space.

To further improve sampling, let us first denote $p(x|r)$ in the latent space of the G-VAE as $p(f|r)$. We follow Liu et al. (2019) and utilize a CCNF formulation (Chen et al., 2018; Abdal et al., 2021) to map $\mathcal{N}$ to $p(f|r)$ by running forward in time from $\vec{u_0} = \epsilon$ to $\vec{u_T} = \vec{u_0} + \int_{t=0}^T g(u_t, t, z(r))dt$, according to the dynamics parameterized by a neural network $g$. By repeating the forward process for different samples $\epsilon$ and a given reference $r$ we can obtain a distribution of generated designs:

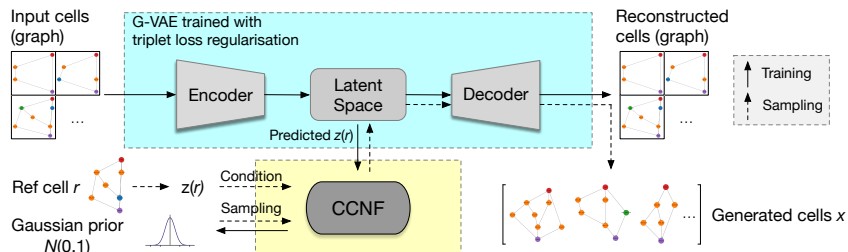

Figure 3: Training and sampling processes of the proposed G-VAE and CCNF models.

| Sampling method | Gen. success | ZC vec. | | | | C-10 Acc. | | |
|---|---|---|---|---|---|---|---|---|
| | | NASWOT | SNIP | Params. (M) | FLOPs (M) | Min | Mean | Max |
| Random | 9/20 | $793_{\pm 49}$ | $104_{\pm 64}$ | $0.3_{\pm 0.3}$ | $111_{\pm 113}$ | 10.00 | $75.00_{\pm 28.37}$ | 93.86 |
| NATS-Bench ref. | | 783 | 80 | 1.15 | 327 | | 94.24 | |
| Neighbourhood | 20/20 | $793_{\pm 49}$ | $92_{\pm 29}$ | $0.7_{\pm 0.3}$ | $220_{\pm 86}$ | 89.76 | $92.15_{\pm 1.13}$ | 94.13 |
| CCNF | 20/20 | $788_{\pm 10}$ | $78_{\pm 13}$ | $0.6_{\pm 0.3}$ | $189_{\pm 100}$ | 91.89 | $93.08_{\pm 0.85}$ | 94.55 |

Table 1: Comparison of different approaches of sampling the G-VAE latent space. Neighbourhood and CCNF are reference-based (see Sec. 3.2) and we test them by giving the best cell from NATS-Bench-TSS (Dong et al., 2021) as a reference. All macro models constructed using $\alpha_x$ from Eq. 1.

$p(u_{\overrightarrow{T}}|r)$. To ensure $p(u_{\overrightarrow{T}}|r)$ approximates $p(f|r)$, parameters of $g$ are fit to minimize negative log-likelihood of generating $f(r)$ from $z(r)$, following the reverse process:

$$\mathcal{L}_{\text{CCNF}} = \mathbb{E}_{p(r)} - \log p(u_T^{\leftarrow}|r) = \mathbb{E}_{p(r)} \Big[ -\log p(u_0^{\leftarrow}) + \int_{t=0}^{T} \text{Tr}\Big(\mathbf{J}_{u_t} g(u_t, t, z(r))\Big) dt \Big], \quad (6)$$

$$\text{where:} \quad u_T^{\leftarrow} = f(r), \ u_0^{\leftarrow} = u_T^{\leftarrow} - \int_{t=0}^{T} g(u_t, t, z(r)) dt, \quad (7)$$

and $\mathbf{J}_{u_t} g$ is Jacobian of $g$ w.r.t. $u_t$. Conditioning on the reference $r$ is done using the same $z(r)$ encoding as in Sec. 3.1 for consistency with the G-VAE. We use Hutchinson's trace estimator to efficiently estimate the trace in Eq. 6 (Grathwohl et al., 2019) and optimize using gradient descent via the adjoint method (Chen et al., 2018). Furthermore, similar to training the G-VAE, to scale training of the CCNF without evaluating $z(r)$ for the excessive amount of models, we utilize the previously trained predictor $q$.

After training finishes, $p(x|r)$ is recovered by simply applying the G-VAE decoder to the samples generated by the CCNF. The overall training and sampling process is summarized in Fig. 3. We validate the quality of designs generated by the CCNF and compare it to naive sampling in the latent space in Tab. 1. We can see that samples from the CCNF are focused closer to the reference compared to naive sampling of the neighbourhood – this results in almost 1 percentage point higher average accuracy of the sampled models and better best- and worst-case performance. Notably, by sampling with a CCNF we are able to find a better design, while neighbourhood sampling fails to achieve this.

### 3.3 MACRO ARCHITECTURE GENERATION

Clustering the micro design space and being able to generate high-performance architectures from each cluster efficiently enables us to factorize the search space $S$ into a two-level hierarchy by replacing individual points from $\Omega$ with identified clusters: $\hat{S} = (\{\hat{\Omega}_k\}_{k=1}^{K} \times C \times \{1, 2\})^{20}$. This in turn reduces the space of macro designs from c. $10^{390}$ to $(K|C||S|)^{20} \approx 10^{78}$. We will refer to $\hat{S}$ as higher-level (HL) search space to distinguish it from the original $S$. Similarly, we will call elements of $\hat{S}$ HL designs/sequences and denote them with $\hat{\alpha}(\hat{t}_1, ..., \hat{t}_{20})$.

Within the HL search space, each triple $\hat{t}$ represents a large number of different possible triples $t$, depending on which design $x$ is selected from a cluster $\hat{x} \in \{\hat{\Omega}_k\}_{k=1}^{K}$. We want to use our CCNF to resolve this, but we face the problem of selecting a reference point – if we already know a strong design point, we can use it and focus our search towards similar designs; this is related to summarizing each cluster with an approximation of $\min_{x \in \hat{x}} \mathcal{L}(\alpha_x)$. However, knowing good designs in advance is a strong assumption. Therefore, we opt for a simpler but more realistic approach of using the most representative models from each cluster as reference – this is conveniently done by conditioning the

CCNF on each cluster's mean $\mu_i$, and is more closely related to considering $\mathbb{E}_{x \in \hat{x}} \mathcal{L}(\alpha_x)$. Formally, for the HL design $\hat{\alpha} \in \hat{S}$, a random design $\alpha(\hat{\alpha}) \in S$ can be defined as:

$$\alpha(\hat{\alpha}) = \alpha(t_1, ..., t_{20}), \quad \text{where:} \quad t_i = [\omega(\hat{x}_i), \hat{c}_i, \hat{s}_i], \tag{8}$$

$$\omega(\hat{x}) = f^{-1}\Big(\epsilon + \int_{t=0}^{T} g(u_t, t, \mu(\hat{x}))\Big), \tag{9}$$

$[\hat{x}_i, \hat{c}_i, \hat{s}_i] = \hat{t}_i$ are elements of HL design $\hat{\alpha}$ and $\mu(\hat{x})$ is the mean of cluster $\hat{x}$.

Having chosen a strategy to sample from each cluster, we then start considering different HL designs for the purpose of training a macro architecture generator. Our goal is to train a sequence generator (SG) $h$ that would propose different parametrized cells in an autoregressive fashion, given a starting condition $y$: $\hat{t}_i = h(..., \hat{t}_{i-1}, y)$. The conditioning token $y$ is similar to the one used for CCNF and guides the generation process towards macro designs with high-level parameters similar to those in $y$. In particular, in our experiments we use $y$ to constrain FLOPs and parameters. We use decoder-only transformer (Radford et al., 2018) model for $h$. Specifically, we fine-tune a pretrained GPT-Neo-125M model provided by Black et al. (2021), by minimizing the negative log-likelihood of generating observed macro sequences from their relevant values of $y$. However, naively training the SG on a random sample of models would result in effectively performing a random search among the models meeting the requested $y$. To mitigate this, we run an evolutionary search (ES) in the HL space (using a CCNF to construct cells) to find designs that maximize T-CET, a state-of-the-art zero-cost proxy (Xiang et al., 2023b), for various size constraints. We then train the SG on the history of the ES, resulting in a bias towards models navigating the trade-off between T-CET and parameters:

$$\mathcal{L}_{\text{SG}} = \mathbb{E}_{\hat{\alpha} \in H} \sum_i -\log p(\hat{t}_i | ..., \hat{t}_{i-1}, y), \quad \text{where:} \quad y = z(\alpha(\hat{\alpha})), \tag{10}$$

$$H \approx \arg\max_{\hat{\alpha}} \text{T-CET}(\alpha(\hat{\alpha})), \tag{11}$$

$\hat{t}_i$ are elements of sequence $\hat{\alpha}$ and Eq. 11 refers to performing the ES (details of the HL-Evo algorithm can be found in App. C). Overall, after the training has finished, the distribution of architectures generated for a given condition $y$ can be summarized as:

$$p(\alpha|y) = p(\alpha(\hat{\alpha})), \quad \text{where} \quad \hat{\alpha} = [\hat{t}_1, ..., \hat{t}_{20}], \quad \text{s.t.} \quad \hat{t}_i = h(..., \hat{t}_{i-1}, y), \tag{12}$$

and randomness is introduced through $\epsilon$ in $\alpha(\hat{\alpha})$ (see Eq. 8-9). Note that we employ deterministic sampling at the macro level.

**Final model selection.** The above sampling procedure is supposed to produce a small, optimized search space for a given conditioning $y$. As such, it should be used with a separate searching algorithm on top of the generated models. Although many approaches could be tried, for simplicity and to keep the cost low, we opted for using a simple selection based on T-CET in our experiments. Different searching strategies are compared in App. E.1.

## 4 EVALUATION

We evaluate our method on the typical datasets: CIFAR-10, CIFAR-100 (Krizhevsky, 2009) and ImageNet-1k (Deng et al., 2009), abbreviated C-10, C-100 and IN-1k, respectively. To test generalizability of our method to less-studied tasks, we also evaluate on the recently-proposed NAS-Bench-360 (NB360, Tu et al. (2022)).

**Pretraining details.** Details about pretraining can be found in App. A. The total pretraining cost is around 30 GPU hours on a single V100 GPU, and the most time consuming part is running the ES for training the SG (14 GPU hours). On the other hand, training the G-VAE and the CCNF takes only 5.5 GPU hours in total. Note that the cost is paid once for all reported experiments.

**Hidden cost.** Additionally to the NAS-specific cost above, our method relies on having access to a pretrained GPT. Although GPTs have become increasingly available, if a model like that had to be trained from scratch (e.g., due to legal issues) it would incur relatively high additional cost. Specifically, Brown et al. (2020) report total compute to train GPT-3 125M to be $2.25 \times 10^{20}$ FLOPs, which translates to approx. 910 GPU hours using a A100 GPU (assuming FP32 with 44% of the peak tensor utilisation, c.f. Tab. 1 by Narayanan et al. (2021)).

### 4.1 SYSTEMATIC COMPARISON ON CIFAR AND IMAGENET

We begin by comparing our method to other approaches based on zero-cost proxies in a controlled setting where all models are trained using exactly the same protocol, architecture constraints and

| Design Space | Method | Pretraining time | ZC evals. | C-10 (1M param.) | | C-100 (1M param.) | | IN-1k (450M FLOPs) | |
|---|---|---|---|---|---|---|---|---|---|
| | | | | Acc.(%) | Cost | Acc.(%) | Cost | Acc.(%) | Cost |
| ZenNet‡ | Evo(Zen) | - | 480,000 | 96.2 | 14 | 80.1 | 14 | - | - |
| | Evo(ZiCo) | - | 480,000 | 97.0 | 10 | 80.2 | 10 | - | - |
| | Evo(T-CET) | - | 480,000 | 97.2 | 19 | 80.4 | 19 | - | - |
| GraphNet | Evo(T-CET) | - | 480,000 | 96.4 | 22 | 79.6 | 22 | 77.6 | 20 |
| | HL-Evo(T-CET)+CCNF | 5 | 20,000 | 97.4 | 3.6 | 80.9 | 3.6 | 78.4 | 6 |
| | SG+CCNF | 30 | 100 | 97.6 | 0.08 | 81.0 | 0.08 | 78.5 | 0.08 |

Table 2: Comparison with SOTA ZC NAS. All methods were run using the same protocol: up to "ZC evals." models were scored using a selected ZC proxy, where model selection was guided with either evolution, HL evolution, or our SG. The top-$N$ models according to each proxy were trained and the best one is reported – for CIFARs, $N$=10, for IN-1k, $N$=1. Constraints denoted under each dataset's name, models that violate these were discarded. "Cost" refers to searching cost in GPU hours.

| Model | Design space | Training scheme | Pre-cost | Search cost | Acc.(%) | Params.(M) | FLOPs(M) | Approach |
|---|---|---|---|---|---|---|---|---|
| OFA (Cai et al., 2020) | MobileNetv3 | Custom | 1,200 | 40 | 79.1 | ? | 389 | OS + predictor |
| OFA scratch (Cai et al., 2020) | MobileNetv3 | Custom | 1,200 | 40 | 77.0 | ? | 389 | OS + predictor |
| OFA scratch (Moons et al., 2021) | MobileNetv3 | EfficientNet | 1,200 | 40 | 77.7 | ? | 389 | OS + predictor |
| GPT NAS (Yu et al., 2023) | Custom | ? | ? | 96 | 79.1 | 110.9 | ? | GPT gen. |
| NSENet† (Ci et al., 2021) | Custom | EfficientNet | ? | c. 4,000 | 77.3 | 7.6 | 333 | SN evo. + OS |
| NSENet-GPU (Ci et al., 2021) | Custom | EfficientNet | ? | c. 4,000 | 77.9 | 15.7 | ? | SN evo. + OS |
| AutoSpace† (Zhou et al., 2021) | Custom | ? | 960 | ? | 77.5 | 5.7 | 380 | SN evo. + OS |
| TE-NAS (Chen et al., 2021) | DARTS | ? | 0 | 4.1 | 75.5 | 5.4 | <600 | ZC + SN |
| Zero-Cost-PT (Xiang et al., 2023a) | Proxyless | Proxyless | 0 | 1.0 | 76.4 | 8.0 | ? | ZC + SN |
| ZenNet (400M-SE) (Lin et al., 2021) | MobileNetv2 | ZenNAS | 0 | 12 | 78.0 | 5.7 | 410 | ZC evo. |
| ZiCo (450M) (Li et al., 2023) | MobileNetv2 | ZenNAS | 0 | 9.6 | 78.1±0.3 | ? | 448 | ZC evo. |
| **Ours** | GraphNet | ZenNAS | 30 | 0.08 | 78.3±0.2 | 4.3 | 429 | ZC gen. + ZC sel. |
| ZiCo (600M) (Li et al., 2023) | MobileNetv2 | ZenNAS | 0 | 9.6 | 79.4±0.3 | ? | 603 | ZC evo. |
| **Ours** | GraphNet | ZenNAS | 30 | 0.08 | 79.5±0.1 | 7.0 | 593 | ZC gen + ZC sel. |
| ZiCo (1000M) (Li et al., 2023) | MobileNetv2 | ZenNAS | 0 | 9.6 | 80.5±0.2 | ? | 1005 | ZC evo. |
| **Ours** | GraphNet | ZenNAS | 30 | 0.08 | 80.6±0.4 | 8.2 | 964 | ZC gen + ZC sel. |

Table 3: Comparison with various NAS approaches on ImageNet-1k for budgets of 450M, 600M and 1000M FLOPs. All numbers other than "Ours" are directly taken from relevant papers. OS – One-Shot, SN – supernet, ZC – zero-cost. Cost in GPU hours.

(max) search budget. We also ablate our method by progressively adding different components and quantifying their effects when performing a search in our search space (which we call **GraphNet**). The results are presented in Tab. 2.

We compare to evolution-based ZC NAS approaches: ZenNAS (Lin et al., 2021), ZiCo (Li et al., 2023) and T-CET (Xiang et al., 2023b). We do not compare to supernet-based ZC NAS, such as TE-NAS (Chen et al., 2021) or Zero-Cost-PT (Xiang et al., 2023a), since constructing a supernet for those methods is impossible in our search space. Overall, we can see that models designed in the GraphNet search space using our proposed generative models achieve superior performance – we attribute this simply to the fact that our search space is much larger, *i.e.*, it is a strict superset of ZenNet design space. However, we can also see that simply increasing the search space size is not enough – when naively using existing methods in our large space, the performance is likely to drop even for a fairly large computational budget (*e.g.*, Evo(T-CET), which is the best on ZenNet, drops by 0.8 percentage points on both C-10 and C-100 when run in the GraphNet space). Compared to these methods, by utilizing our G-VAE and CCNF and running evolution in the HL space, we can already improve upon all existing methods while being significantly cheaper (HL-Evo) – this is primarily due to the informative organization of the search space (Sec. 3.1). Finally, we can amortize the cost of running the HL ES each time by training our SG, which then allows us to achieve comparable, if not slightly better, performance in a matter of minutes – this property is desired since it allows us to mitigate the "cold start" problem in NAS, addressed by some recent work (Zhao et al., 2023). More experiments using different algorithms on our GraphNet space can be found in App. E.2.

## 4.2 OPEN-WORLD COMPARISON ON IMAGENET

Here we compare to other NAS approaches that can be found in the literature without requiring them to run in an aligned setting – the only common trait is the models should be (roughly) smaller than 450M FLOPs. Our focus is on positioning our work w.r.t. the state-of-the-art approaches in the 3 most relevant lines of work: *1)* zero-cost and low-cost NAS, *2)* automated search space design, and *3)* generative NAS. We present detailed comparison in Tab. 3, including our best effort to highlight any differences in search spaces and training schemes. We can see that our method achieves very strong

| Method | Spherical | Darcy Flow | PSICOV | Cosmic | NinaPro | ECG | Satellite | DeepSEA |
|---|---|---|---|---|---|---|---|---|
| Expert | **67.41** | 0.008 | 3.35 | **0.13** | 8.73 | **0.28** | 19.8 | 0.30 |
| WRN Zagoruyko & Komodakis (2016) | 85.77 | 0.073 | 3.84 | 0.24 | 6.78 | 0.43 | 15.49 | 0.40 |
| DASH Shen et al. (2022) | 71.28 | **0.008** | 3.30 | 0.19 | **6.60** | 0.32 | **12.28** | **0.28** |
| **Ours** | 71.03 | 0.014 | **3.13** | 0.14 | 6.90 | 0.32 | 18.11 | 0.50 |

Table 4: Results on the tasks from NB360. Metrics and baselines follow what is reported by Shen et al. (2022). Lower is better for all tasks. Best in bold, second best underlined. For the ECG task, we consider Ours to be better than DASH due to the lower searching cost.

performance while requiring minimal computation resources. The only two methods that achieve better results are OFA (when initialized with weights from the supernet) and GPT NAS. However, the former requires extensive pretraining while the latter is almost $33\times$ larger (the authors do not report FLOPs so we included the model size, but it probably violates the FLOPs constraint as well). Results of our method are averaged from searching 3 times, each time training the resulting network once.

### 4.3 NAS-BENCH-360

To run on NB360, we first have to decide on a sensible conditioning for our SG. Unfortunately, unlike the previous benchmarks, NB360 does not have well-established constraints that we could use and compare to other methods with a similar budget — to resolve this, we decided to use #params and #FLOPs of the Wide-ResNet (WRN) as this is the most basic baseline considered by Tu et al. (2022). Similar to the above, we generate 10 architectures, use T-CET to select the best, then train it. We compare to DASH (Shen et al., 2022), which is a recent NAS method designed specifically with NB360 in mind, and expert-designed networks in Tab. 4. We can see our generative approach achieves very competitive performance. Importantly, we are able to improve upon the WRN reference in 5/8 cases. What is more, we improve upon DASH in 4/8 cases, despite DASH performing supernet training and HPO for each task, whereas we do not use any feedback. Note that due to amortised cost of training the SG we are also much faster — generating models for new tasks take minimal time. Still, there is room for improvement, considering our method fails noticeably on DeepSEA and Satellite tasks. We leave investigating those failure cases for future work.

## 5 LIMITATIONS, DISCUSSION AND FUTURE WORK

**Reliance on ZC proxies.** Our method heavily relies on ZC proxies – we do that in order to organise a large search space like ours in a meaningful way without incurring a prohibitively large comp. cost; to the best of our knowledge, there is no better alternative at the moment. Having said that, ZC proxies are known to not always produce faithful scoring of networks (White et al., 2021; Ning et al., 2021). Although we try to avoid the most common failure cases by relying on clustering rather than strict ordering, our assumptions might still not hold in general. On the other hand, due to the paradigm shift in our work, we show a new direction for future research on ZC proxies focused on relative performance – our work shows that this approach can be more powerful even in settings where existing proxies are known to work well (e.g., CIFAR). Another possible direction would be to incorporate feedback mechanism to efficiently adjust network generation in an online fashion.

**Conditioning.** Currently micro cells are generated based on cluster centers – this is simple and shown to work well in our case, but in the end leaves room for improvement. The method could also be extended to include more explicit conditioning on the target task (currently #FLOPs and #params).

**On-device performance.** On-device performance is important for the practical deployment of models, but we did not consider it a direct objective for simplicity, instead relying on device-agnostic metrics such as a number of parameters and FLOPs. Extending our work to be able to design networks optimized for a particular device is a very sensible direction for future work.

**Transformers.** Although transformers keep playing increasingly important role, for the time being we do not consider them in our search space. Extending our codebase to support hybrid convolution-attention models is an exciting direction for future work.

## 6 CONCLUSION

We have presented a novel method to efficiently generate high-performance DNN architectures for user-defined architectural constraints, by using an elaborated hierarchy of generative models, guided by the clustering properties of zero-cost proxies. Our approach achieves state-of-the-art performance on common tasks, including avg. 78.3%, 79.5% and 80.6% top-1 accuracy on ImageNet-1k for the budgets of 450M, 600M and 1000M FLOPs. We hope our work serves as a proof of concept that generative methods can be successfully used for NAS without incurring prohibitive cost.

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

## A   PRETRAINING: ARCHITECTURES, HYPERPARAMETERS AND COST

In this section, we furnish an overview of the hyper-parameters employed in each component, along with the training parameters utilized during their respective training procedures. We also provide detailed overview of the cost of different steps of our method in Tab. 5.

The G-VAE is configured with four graph convolution layers (Kipf & Welling, 2017), utilizing a hidden state of size 512 and projecting to an output latent space of size 256 for the metrics predictor. This metrics predictor is structured as a two-layer MLP with ReLU activation. The first layer serves as a hidden state, mirroring the size of the latent space at 256, and the subsequent layer produces four conditions, aligning with our target outcomes.

In terms of training, the metrics predictor undergoes joint training with the G-VAE. The AdamW optimizer Loshchilov & Hutter (2019), combined with a cosine annealing schedule (Loshchilov & Hutter, 2017), is employed, initiating with a learning rate of 1e-3 and decaying to a minimum value of 0. A weight decay is also incorporated, set at 5e-4. The training procedure is constrained by a maximum of 500,000 steps and incorporates an early stopping mechanism. This mechanism ceases training if a reduction in loss is not observed over a span of 10 epochs.

The Continuous Conditional Normalizing Flow (CCNF) incorporates nine Concat+Squash+Linear layers, each having a hidden dimension of 512. The input layer is structured to accommodate a latent feature size of 256 and is engineered to project these latent features into a Gaussian distribution, preserving the identical dimensional size of 256.

For training, we utilize 400,000 unlabeled graphs as our dataset. These graphs generate latent features through G-VAE, which are further predicted into ZC vectors by the metrics predictor. The CCNF is trained with a fixed learning rate of 1e-3 and a batch size of 256. Weight decay is set at 0.01, serving as the default value for the Adam optimizer (Kingma & Ba, 2015). The latent features are then transformed into a standard Gaussian distribution, which acts as the prior distribution by minimizing log-likelihood. Notably, the network employs the "dopri5" ODE solver (Hairer et al., 1993) with both absolute and relative tolerance set to 1e-5.

For training the sequence generator, GPT-Neo-125M (Black et al., 2021) is employed as the foundation through pre-trained checkpoints. The model consists of 12 transformer blocks (Vaswani et al., 2017) using GELU (Hendrycks & Gimpel, 2016) and LayerNorm (Ba et al., 2016). We fine-tune this model with a consistent learning rate of 1e-3, utilizing the 60,000 networks and zc-vectors gathered from the HL-Evo procedure and maintaining a batch size of 1. To augment the dataset and mitigate the risk of overfitting, we enhance our condition token by randomly dropping tokens from the set that includes Param, Flops, and ZC values.

For details of our hyperparameters, please refer to Tab. 6.

|  | G-VAE | Metrics Predictor | CCNF | Seq. Gen. |
|---|---|---|---|---|
| Data gen. | ——————— 5 ——————— |  |  | 14 |
| Training | ——— 0.5 ——— |  | 0.5 | 11 |

Table 5: Detailed cost of running each step of our method All cost in GPU hours assuming execution on a V100 GPU.

## B   OPERATIONS INCLUDED IN THE DESIGN SAPCE

This section introduces the foundational operations incorporated in our graph design space. Our design space consists of nine basic operations, listed as follows:

- **avg_pool**: Average Pooling is a downsampling technique utilized to reduce the spatial dimensions of a feature map. It calculates the average value of the elements within a specified local region determined by the kernel size and stride.

- **conv**: The Convolutional Layer starts with a ReLU pre-activation and ends with a batch normalization (Ioffe & Szegedy, 2015) layer. This operation is pivotal for learning spatial hierarchies of features and is widely employed in deep learning models.

|  |  | G-VAE | Metrics Predictor | CCNF | Seq. Gen. |
|---|---|---|---|---|---|
| **ARCH** | Main Operation | GCN | Linear | Cat + Squash + Linear | Transformer |
|  | Number of layers | 4 | 2 | 9 | 12 |
|  | Input dimension | $[9 \times 9, 15 \times 9]$ | 256 | 256 | 64 |
|  | Hidden dimension | 512 | 256 | 512 | 2048 |
|  | Output dimension | 256 | 4 | 256 | 768 |
|  | Activation fn | ReLU | ReLU | tanh | GELU |
|  | Normalisation | LayerNorm | - | Moving BatchNorm | LayerNorm |
|  | Prior | Gaussian | - | Gaussian | - |
| **TRAIN** | Tr. samples | 400K unlabelled + 50k labelled | | 400k | 60k |
|  | Batch size | 64 | | 256 | 1 |
|  | Learning rate | 3.5e4 | | 1e3 | 1e3 |
|  | Learning rate schedule | CosineAnnealingLR | | fixed | fixed |
|  | Weight decay | 5e-4 | | 0.01 | 0.05 |
|  | Optimizer | AdamW | | Adam | Adam |
|  | Loss | Triplet + BCE + KLD + MSE | | $\log p$ | CE |
|  | Augmentations | - | | - | randomly drop condition tokens from set of [param, flops zc ] |
|  | Others | Trained jontly | | solver='dopri5' atol=1e-5 rtol=1e-5 | GPT-neo-125 |

Table 6: Detailed hyperparameters of each component.

- **sep_conv** (Niklaus et al., 2017): Separable Convolution is an efficient variant of the standard convolution. It factorizes a standard convolution into a depthwise spatial convolution followed by a pointwise convolution, reducing the computational costs.

- **dil_conv** (Yu & Koltun, 2016): Dilated Convolution introduces gaps to the convolutional kernel, effectively increasing the receptive field without increasing the number of parameters, making it suitable for tasks requiring the incorporation of larger contextual information.

- **mbconv** (Yu & Koltun, 2016): The MobileNetV2-inverted bottleneck convolution has three variants with expand sizes at [2, 3, 6]. It employs an inverted residual structure, aiding in building efficient and compact models suitable for mobile applications.

- **bconv** (He et al., 2016): The Bottleneck Convolution comes in two types with shrink rates in the bottleneck at [0.5, 0.25]. It utilizes a bottleneck structure to reduce dimensionality, thereby decreasing computational requirements.

In addition to the basic operation types introduced above, we also offer three choices of kernel sizes – {1, 3, 5} – for each operation. This variety ensures coverage of the most prevalent network design paradigms, allowing for versatile adaptability and exploration within our graph design space.

## C DETAILS OF OUR HL-EVO ALGORITHM

Alg. 1 summarizes our HL evolutionary search algorithm (HL-EVO), which combines evolutionary search with the pretrained CCNF to discover good-performing macro architectures. Our HL-EVO consider only mutations – parents are randomly selected, while the offsprings with low fitness are dropped. In practice, mutations may occasionally produce child architectures that do not satisfy the pre-defined constrains – they are simply discarded by the algorithm, which then moves forward to the next iteration. In particular, our HL-EVO operates in two phases, *exploration* and *exploitation*, with equal budget on number of iterations. In the exploration phase, we allow the search to mutate across different cell families for each $\hat{x}_i \in \hat{\alpha}(\hat{t}_1, ..., \hat{t}_{20})$, so it can explore various combinations of cell families in the macro sequence. On the other hand, in the later exploitation phase, we restrict the search to fix the current sequence of cell families $\hat{x}_i$, but only mutate channel $\hat{c}_i$ and stride $\hat{s}_i$, in hope to keep the already explored promising macro sequences, while relying on the CCNF to subsequently exploit good-performing final architectures from these macro sequences $\alpha(\hat{\alpha})$.

---

**Algorithm 1:** HL-Evo

---

**Input :** Search space $S$ factorized into HL space $\hat{S}$ and clusters $\{\hat{\Omega}_k\}_{k=1}^K$, objective $\mathcal{L}$, total number of iteration $T$, population size $N$, initial architecture $\alpha_0$, a trained CCNF model $g$ and a G-VAE decoder $f^{-1}$

**Result:** A selected architecture $\alpha^\star$, search history $H$

1 **Function** GenCell($\hat{\Omega}$)**:**
2     $\epsilon \sim \mathcal{N}$
3     $x \leftarrow f^{-1}\big(\epsilon + \int_{t=0}^{T} g(u_t, t, \mu(\hat{\Omega}))dt\big)$
4     **return** $x$
5 **End Function**

6 **Function** Mutate($\alpha$, *exploration*)**:**
7     take random cell $t = [x, c, s] \in \alpha$
8     let $\hat{x}$ be the cluster to which $x$ belongs
9     **switch** *random mutation* **do**
10       **when** change channels $\Rightarrow$ sample new $c \sim \mathcal{U}\{C\}$
11       **when** change stride $\Rightarrow$ sample new $s \sim \mathcal{U}\{1, 2\}$
12       **when** change cell design $\Rightarrow$ generate new $x \leftarrow$ GenCell($\hat{x}$)
13       **when** exploration && change cell type $\Rightarrow$
14          sample new cluster $\hat{x} \sim \mathcal{U}\{\hat{\Omega}_k\}$,
15          then generate new cell $x \leftarrow$ GenCell($\hat{x}$)
16     **end**
17     replace $t$ in $\alpha$ with its modified version
18     **return** modified $\alpha$
19 **End Function**

20 $H = \{\alpha_0\}$
21 $P \leftarrow H$
22 **for** $e \in \{$true, false$\}$ **do**
23     **for** $i \leftarrow 0$ **to** $T/2$ **do**
24       take random model from the population $\alpha \sim \mathcal{U}\{P\}$
25       $\alpha' =$ Mutate($\alpha, e$)
26       **if** $|P| \geq N$ **then**
27          $\alpha'' \leftarrow \arg\max_{x \in P} \mathcal{L}(x)$
28          $P \leftarrow P \setminus \{\alpha''\}$
29       **end**
30       $P \leftarrow P \cup \{\alpha'\}$
31       $H \leftarrow H \cup \{\alpha'\}$
32     **end**
33 **end**
34 $\alpha^\star \leftarrow \arg\min_{x \in H} \mathcal{L}(x)$

---

In our experiments, we use $T = 20000$ and $N = 512$. We choose a relatively small initial architecture $\alpha_0$ as $[(\hat{x}^{(5)}, 32, 1), (\hat{x}^{(5)}, 64, 2), (\hat{x}^{(5)}, 64, 1)]$, where $\hat{x}^{(5)}$ is the mean of the 5-th cluster (cell family).

## D    CODE

Code will be available upon acceptance.

## E    MORE EXPERIMENTS

### E.1    ALTERNATIVE FINAL SEARCH ALGORITHM

As mentioned at the end of Section 3, after our method generates a small number of promising models, there remains the question of the final selection. Although we opted for simple selection using T-CET in our main experiments, other approaches are also possible, although they are likely to increase the overall cost. To further explore this direction, we also include an approach of running a typical baseline — evolutionary search with reduced-training proxy Zhou et al. (2020). For this purpose, we utilised training scheme similar to the one used by NATS-Bench-201 Dong et al. (2021), with batch size 256 and 50 epochs on CIFAR-100. Using this scheme, training a single model

takes approx. 0.5h, so we are able to train 40 models for a common budget of 20h presented in Tab. 2. Given the budget of 40 models, we run evolution using a population size of 8, to allow for a reasonable number of mutations. We run this baseline on the optimised search space obtained from generating 5 models from the same macro sequence and mixing different instances of cells between the 5 models. In our experiment, this resulted in the space of $5^4$ models. The best model according to the reduced-training proxy was trained fully using the same training procedure as the rest of methods in Tab. 2. The final performance turned out to be 97.3 — this is somewhat noticeably lower than selection with T-CET (97.6), especially considering much higher searching cost, but also much higher compared to using a similar reduced-training proxy directly on the large GraphNet space (see below). In general, the results suggest our SG+CCNF can indeed help a searching a lot by shortlisting promising architectures, although it is probably not the best idea to use reduced training to select the final model.

### E.2 MORE BASELINES ON THE BIG GRAPHNET SPACE

In section 4 we show that Evo(T-CET) on the GraphNet space achieves inferior performance. But what about other searching strategies? Here we again include a common baseline of running standard evolutionary algorithm with the same reduced-training proxy as above. Given the same searching budget of around 20 GPU hours, this baseline achieved accuracy of 90.32, a significant shortcoming compared to 96.4 of Evo(T-CET) and 97.6 of our SG+CCNF. This showcases challenges of using popular off-the-shelf black-box optimiser in very large search spaces like ours.

