# OpenReview forum: "Towards Neural Architecture Search through Hierarchical Generative Modeling"
_ICLR.cc/2024/Conference — ICLR 2024 Conference Withdrawn Submission_

### Official Review · Reviewer_2D3t · 2023-10-20

**Soundness:** 3 good
**Presentation:** 3 good
**Contribution:** 3 good
**Rating:** 6
**Confidence:** 4

**Summary:**

The authors propose an efficient method of constructing and searching through neural architecture search spaces. Their method consists of multiple stages. In the first stage, a graph variational autoencoder is trained to produce embeddings of different architectures on the cell level. The authors add a margin loss comparing architectures using zero-cost proxies to encourage the architectures to cluster. The resulting latent space is clustered using a gaussian mixture model.
In order to sample architectures given a well-performing reference architecture in the high dimensional latent space of the VAE, the authors employ a conditional, continuous normalizing flow model. In order to create the macro architectures, the authors propose to fine-tune a decoder-only generative transformer model on architectures found via evolutionary optimization, which uses a zero-cost proxy for its search.
The authors demonstrate competitive performance with both aligned methods on CIFAR-10 and CIFAR-100 as well as overall NAS methods on Imagenet.

**Strengths:**

- The method is well motivated, each problem encountered in the architecture generation process is clearly explained and their approach to solve it makes sense to me.
- The authors openly state the limitations and assumptions made by their framework, I appreciate the level of detail.
- Overall clearly written paper.

**Weaknesses:**

- CIFAR-10, CIFAR-100 and Imagenet are benchmarks which the community has collectively overfit on. If time permits do you think you could provide results on a different modality, e.g. an NLP task?
- To better understand the variance of your method, could you rerun the architecture selection multiple times and also report conf. intervals over the variance of each architecture when run for 3-5 random initializations?
- As stated in their limitations, the method largely relies on ZC proxies for both G-VAE and the generation of the macro architecture.
- Code was not submitted alongside the submission.

**Questions:**

- How do you estimate the number of components for the GMM?
- Why is the cost of the ES for training the SG such a major bottleneck when using ZC proxies? Did you parallelize ES?


Typos (only minor, just listing them for completeness):
- 'even the best searching algorithm' -> 'even the best search algorithm'
- '... impressive results for modelling highly-dimensional conditional ...' -> '... impressive results for modelling high-dimensional conditional ...'
- 'to a continues latent space' -> 'to a continuous latent space'

---

> ### Author Response · Authors · 2023-11-23
> **Thank you for your positive feedback.**
>
> We are deeply grateful for your positive evaluation and insightful comments on our manuscript. Your feedback not only encourages us but also enhances the quality of our research. Regarding the issues you have raised, we hope the following answers are satisfied. Please let us know if you see any further issues in the paper that are unclear or need to be addressed.
>
>
> > If time permits do you think you could provide results on a different modality, e.g. an NLP task?We
>
> Since generalizability was a common concern among reviewers, we decided to include experiments on NB360. We hope this should be more than enough to satisfy this request. Please see the common reply summarising new experiments.
>
> >To better understand the variance of your method, could you rerun the architecture selection multiple times and also report conf. intervals over the variance of each architecture when run for 3-5 random initializations?
>
> Please see the common reply summarising new experiments.
>
> >As stated in their limitations, the method largely relies on ZC proxies for both G-VAE and the generation of the macro architecture.
>
> Recent work in zero-cost metrics has shown that zero-cost metrics-based NAS can produce promising performance networks compared with the training-based NAS approach. Admittedly, the zero-cost metrics performance can vary when ranking models from different tasks and different search spaces. However, in our generator, we are not strictly based on the assumption that model performance ranking strictly follows the zero-cost metrics ranking, but ‘similar zero-cost metrics will lead to relatively similar performance’. This is different from existing zero-cost metrics NAS.
>
> >How do you estimate the number of components for the GMM?
>
> To decide the number of components for the GMM we increase their number, starting from 16, and calculate the Bayesian Information Criterion (BIC) and Akaike Information Criterion (AIC). We stop when they plateau.
>
> >Why is the cost of the ES for training the SG such a major bottleneck when using ZC proxies? Did you parallelize ES?
>
> Please note we report GPU-hours, i.e., cost and not wallclock time — to avoid confusion, we will change the term used in the paper accordingly. This cost is subject to parallelization, following common practices. Since all steps can be parallelized trivially, we can achieve almost linear speedup if we use multiple GPUs, e.g., with 4 GPUs we can finish everything within ~8h.
>
> Regarding the relative cost of ES to the rest of the system. The bulk of the cost comes from calculating proxies for models, so it is going to be directly proportional to the amount of “labelled examples” mentioned in Tab. 5 (compared to calculating proxies, operating on graphs is very fast, e.g., when training a G-VAE or running a metrics predictor). As such, getting the training data for G-VAE and the SG should take similar time (50k vs 60k), but note that the micro model uses only up to 64 channels (Eq. 2) and our macro search space allows for <=1024, so the average model is much larger in the macro stage.
>
> Finally, we did not dedicate a lot of time optimising our code and there are some opportunities to achieve non-negligible speedups. To give an example, when computing the vector of proxies we currently compute each of: NASWOT, SNIP-SSNR and FLOPs completely independently from each other, resulting in 3 forward passes and 1 backward pass per model, but it clearly should be possible to optimise it such that only 1 forward and 1 backward is needed per model. This would reduce the time needed for data generation by up to 2x.

---

### Official Review · Reviewer_sRLa · 2023-10-26

**Soundness:** 2 fair
**Presentation:** 3 good
**Contribution:** 2 fair
**Rating:** 3
**Confidence:** 4

**Summary:**

This paper aims to reduce the dependency of NAS on the search space design to improve NAS's applicability to less-studied tasks. The core technique is a new hierarchical generative model pretrained using a metric space of graphs and their zero-cost similarity. Experiments are conducted on several standard benchmark datasets, including CIFAR and ImageNet.

**Strengths:**

1. This paper is well-motivated.
2. Improving the applicability of NAS to less-studied tasks is a fundamental challenge for NAS and has large practical values.

**Weaknesses:**

1. The proposed method still heavily depends on human prior knowledge (e.g., reference design, zero-cost metric). I do not see any improvements in improving NAS's applicability to less studied domains.
2. The generalization of the proposed method is a big question. As far as I know, the generalization ability of zero-cost metrics is a bit limited. As the proposed method is based on zero-cost metrics, I think it is necessary to verify the proposed method's generalization ability.
3. The idea of using a hierarchical design to reduce the space is not novel.
4. I feel the current experiment design is not a good fit for this paper. I am more interested in seeing experiments on less studied domains instead of these standard benchmark datasets. If the author can show one practical case where their method can clearly outperform conventional NAS, this paper will be much stronger.

**Questions:**

Please check my comments above.

---

> ### Author Response · Authors · 2023-11-23
> **Thank you for your feedback.**
>
> Thank you for dedicating your time and review our manuscript. We appreciate your comments and acknowledge the concerns you have raised, we want to make further clarification as follows:
>
> > The proposed method still heavily depends on human prior knowledge (e.g., reference design, zero-cost metric)
>
> Regarding the reference design, we would like to point out that in our experiments we used cluster centres for that, which are decided automatically and do not require any prior knowledge. Although we discussed the possibility of improving results further if a good reference point was known, we did not rely on that when running our method. Therefore, we believe the reviewer’s point is reaching too far in that aspect.
>
> Regarding zero-cost proxies, we agree we rely on them, which involves certain risks, as mentioned in the limitations. Still, the proxies are a way of probing into the characteristics of neural networks, like #params or #flops; if anything, we would consider our approach of clustering cell designs according to ZC proxies to be rather automated and require very little human input. We strongly disagree with the implied statement that building on top of the existing knowledge should be considered a weakness.
>
> Having said that, we understand the reviewer is also making a point about the applicability of our method to less-studied tasks. While we disagree with the points above, we do agree with concerns about generalizability, considering our original submission focused on well-studied tasks only. To address this, we have run our method on a suite of tasks from NB360, please find the details in the common reply summarising new experiments.
>
> >The generalization of the proposed method is a big question.
>
> Please see the results on NB360 in the common reply.
>
> >The idea of using a hierarchical design to reduce the space is not novel.
>
> We never claim the idea of using a hierarchical design is novel on its own — if the reviewer got an impression that’s the case, please let us know which part of the paper made them think so and we will revise it. The novelty of the paper lies in designing a hierarchical generative system, which is the first of its kind, to the best of our knowledge.
>
> >I feel the current experiment design is not a good fit for this paper. I am more interested in seeing experiments on less studied domains instead of these standard benchmark datasets.
>
> Please see the results on NB360 in the common reply.

---

### Official Review · Reviewer_LpBV · 2023-10-30

**Soundness:** 3 good
**Presentation:** 3 good
**Contribution:** 2 fair
**Rating:** 5
**Confidence:** 4

**Summary:**

The paper studies a zero-cost NAS that navigates an extremely large, general-purpose search space to produce network of good quality. The proposed schema considers micro level cell designs at first, then leverages a transformer generator to produce macro architectures for a given task and architectural constraints. Numerical experiments on CIFAR and ImageNet validate the efficacy of the method.

**Strengths:**

- The paper is written well and technically sound.
- The topic of using generative model into ZC NAS is interesting.

**Weaknesses:**

- The design conflicts the target pain point. The target pain point raised in the introduction is to eliminate the need of designing search space manually. However, this paper still manually designs a search space and develop an algorithm upon it, see page 4.

- The proposed method seems not generic and time-consuming. The proposed methods require a transformer generator to produce macro architectures which seems require time-consuming pretraining, fine-tuning, and may be task and search space specific.

**Questions:**

See the weakness.

---

> ### Author Response · Authors · 2023-11-23
> **Thank you for your feedback.**
>
> Thank you for the time and effort you have invested in reviewing our manuscript. We greatly appreciate your insights and acknowledge the points you have raised. Regarding your concerns, we have provided additional clarification as follows and we are committed to enhancing our experiments and manuscript, hope that the revision will satisfy your concern.
>
> > The design conflicts the target pain point.
>
> We apologise for not making our point sufficiently clear — this is a side effect of limited space, so please let us explain our reasoning in more detail.
>
> First, it is impossible to have an operational codebase that would be able to train models without defining its ramifications. This means any NAS system requires a search space that is designed manually — even if this search space is implicit — because of the need to write functioning code that would be able to execute networks. This is a fundamental issue and we do not see a way around it. We want to make this point clear because the way in which the reviewer raised their point (“this paper still manually designs a search space and develops an algorithm upon it”) directly touches upon it; there is no way to perform NAS differently than designing a search space and developing an algorithm upon it, any deviations from this rule are likely to be nothing more than semantic nuances (e.g., what exactly is called a search space).
>
> Instead, what is usually understood behind the term “manually designed” search space is the amount of manual effort that is needed to construct a search space in which good models can be found fast enough. The trade-off between search space size and time needed to search is again fundamental — we can easily think about a search space of all possible models that can be expressed in, e.g., Pytorch, but searching in such a search space is going to be extremely difficult. Consequently, every NAS method performs some kind of regularisation of this huge search space by considering only its subset, to make the problem tractable. The question becomes: how big can the subset be and if there are any constraining factors. This leaves us with a spectrum, where a method can be considered more “automated” if it retains strong searching performance without limiting the number or the qualitative properties of models in the search space. We would argue that our method advances in both aspects. On one hand, our search space is extremely expressive, to the best of our knowledge the largest among any works that achieve competitive ImageNet performance. On the other hand, we do not impose constraints such as the ability to construct a supernet (with the size of our search space, we would run out of GPU memory very fast if we tried that), etc. As such, the effort to design our search space is rather minimal — we simply include as many operations from the literature as possible, expose as many parameters (channels, stride) as possible and only make sure any network can be executed correctly. Considering all that and the fact that we can obtain strong performance fast, we would argue it is justified to say our method is “more automated” than the existing ones and that it helps minimise manual effort (we will revise our paper to make it clear it is not our goal to completely eliminate manual design, since this cannot be done).

---

> > ### Author Response · Authors · 2023-11-23
> > **Futher Comments**
> >
> > >The proposed method seems not generic and time-consuming. The proposed methods require a transformer generator to produce macro architectures which seems require time-consuming pretraining, fine-tuning, and may be task and search space specific.
> >
> >
> > Regarding genericness, we admit our initial experiments did not necessarily make a strong point behind this aspect. However, with the addition of NB360, we hope this concern is now alleviated.
> >
> > Regarding time, in our opinion, the amortised cost of 30 GPUh puts us in the regime of very efficient NAS methods. We are not sure why the reviewer says it seems time-consuming. Sure, it might not be the very fastest NAS method but it is still very efficient, especially if we consider the size of our search space. In case this was missed, please note: 1) it is GPU hours, not days (more commonly reported), 2) this is further subject to parallelization (every step is trivially parallelizable) - with 4 GPUs we can finish everything within ~8 hours.
> >
> > The only really time-consuming element could be the mentioned pretraining of the GPT model, but we think it is somewhat questionable if this should be considered a direct cost of our method (see our response to reviewer vDUp for more details).
> >
> > If we disregard the cost of pretraining the GPT (since it is not related to NAS in any way) and consider updated results, it is clear that our method is very efficient and not exactly task-specific. With a single pre-training costing us 30 GPU hours, we are able to conduct 13 different searches (C-10, C-100, IN-1k with 3 budgets + 8 tasks from NB360), advancing state-of-the-art on well-established tasks and matching/advancing on 4/8 understudied tasks, all at the same time. To the best of our knowledge, this is a clear advancement compared to the existing methods.

---

### Official Review · Reviewer_vDUp · 2023-11-02

**Soundness:** 3 good
**Presentation:** 3 good
**Contribution:** 3 good
**Rating:** 5
**Confidence:** 4

**Summary:**

This paper proposed efficient hierarchical generative modelling for neural architecture search using zero-cost proxies. NAS as a field often relies strongly on well-designed search spaces. Design of such search spaces is non-trivial especially in new domains and research areas. This paper addresses this disadvantage by  exploiting an extremely large, general-purpose search space efficiently, by training a two-level
hierarchy of generative models. First level of the conditional generative process focusses on micro-cell design using conditional continuous normalizing flow and the second level uses an transformer to sequentially model the macro architecture. What makes this approach effective in these larger spaces is the ability to exploit task-agnostic zero cost proxy scores to pre-train the generative model. The method is evaluated on the cifar10, cifar100 and the imagenet1-k dataset.

**Strengths:**

Originality: I find the main contribution of the paper of modelling a 2-level hierarchical search space (which is very expressive) novel and interesting. Furthermore this paper effectively avoids the expensive training cost of generative nas models by relying on zero-cost proxies for pre-training

Clarity: I found the presentation clear in most parts (refer to questions for things that are unclear)

Significance: This paper in my opinion proposes an interesting way to exploit zero cost proxies to search in a large and expressive search spaces. The results in my opinion are competitive and significant

**Weaknesses:**

- Given the search space design and inability to construct a supernet, fair comparison with effective nas strategies becomes challenging in this case. For example the OFA search space in table 3 is very different from the search space of GraphNet. Hence it becomes difficult to understand if the gains are attributed to search space design itself or the NAS approach. I recommend comparison with black box nas methods on exactly the GraphNet space(eg: regularized evolution [1], hierarchical nas [2])

- The method uses a pretrained GPT-Neo-125M model. Hence the actual search cost also implicitly inclues the pre-training cost of this model, which should ideally be added to the search cost computation (table 1,2,3).

- Study is limited to convolutional spaces. It becomes natural to question the robustness of the proxies to the recent transformer baeed search spaces in NAS

- Performance on Imagenet is still dominated by methods like Once-For-All which models a simple chain structured space  in Table-3 (contrary to the more expressive space here).

- I encourage the authors to release code to foster reproducibility in NAS

[1] Real, E., Aggarwal, A., Huang, Y. and Le, Q.V., 2019, July. Regularized evolution for image classifier architecture search. In Proceedings of the aaai conference on artificial intelligence (Vol. 33, No. 01, pp. 4780-4789).

[2] Schrodi, S., Stoll, D., Ru, B., Sukthanker, R., Brox, T. and Hutter, F., 2022. Towards discovering neural architectures from scratch. arXiv preprint arXiv:2211.01842.

**Questions:**

- Refer points in weaknesses

- Could the search space design and search methodology be extended to transformer spaces like AutoFormer [1] or HAT [2]?

- Could the authors ablate the choice of T-CET as a proxy in macro architecture generation across different proxy choices? How robust/sensitive is the search to different choices of (strong) zero-cost proxies?

- Could the authors ablate the gains from each of the two phases of hierarchical generative modelling? ie fixing the cell and only performing macro search and vice-versa.

- Could the authors study insights derived from NAS method? Which architectural designs tend to be more impactful than the others?

[1] Chen, M., Peng, H., Fu, J. and Ling, H., 2021. Autoformer: Searching transformers for visual recognition. In Proceedings of the IEEE/CVF international conference on computer vision (pp. 12270-12280).

[2] Wang, H., Wu, Z., Liu, Z., Cai, H., Zhu, L., Gan, C. and Han, S., 2020. Hat: Hardware-aware transformers for efficient natural language processing. arXiv preprint arXiv:2005.14187.

---

> ### Author Response · Authors · 2023-11-23
> **Thank you for your feedback.**
>
> We thank the reviewer for the valuable and constructive comments on our work. We hope to have answered all of your questions satisfactorily below. Please let us know if you see any further issues in the paper that are unclear or need to be addressed.
>
> > I recommend comparison with black box nas methods on exactly the GraphNet space(eg: regularized evolution [1], hierarchical nas [2])
>
>
> We have reported the performance of regularized evolution with T-CET in our GraphNet space in Tab. 2 (c.f. Evo(T-CET)). As suggested by the reviewer, we ran additional experiments using regularized evolution with a reduced training scheme (50 epochs training, same as NAS-Bench-201). We refer to this new baseline as Evo(reduced-training) in the following. Under a similar search budget with Evo(T-CET) (~20 GPU hours), we were able to train ~40 models using Evo(reduced-training), and the results are shown in the table below.
>
> To further demonstrate the effectiveness of our approach, in addition to running black box NAS methods directly on the GraphNet space, we also conducted experiments where we run Evo(reduced-training) on an optimised search space produced by our method (SG+CCNF+Evo(reduced-training) in the following table), under a similar search budget (~20 GPU hours). In this setting, we first use our sequence generator (SG) to generate the macro structure while for each cell we use CCNF to propose 5 candidates. We then construct a search space for running Evo(reduced-training) by considering any combination of the 5 generated cells (note: the macro models used 4 cells in this case). In total, the optimised search space is of size 5^4.
>
> We observe that under a similar search budget, regardless of requiring training or not, regularized evolution, i.e. both Evo(reduced-training) and Evo(T-CET), struggle to find good performing models on the large GraphNet space. On the other hand, when equipped with our hierarchical generative approach, the optimized search space significantly bolsters performance of both the standard and the zero-cost evolution.
> Interestingly, SG+CCNF+ZCNAS achieved a bit higher result in this experiment, compared to running reduced training in the optimized space. This should not be a surprise, however, if we recall that reduced training is known to be a somewhat bad proxy, and T-CET is known to be especially good for datasets like CIFAR-10. Still, reduced training did reasonably well, and much better than when running in a full space; we include it for the sake of completeness. We also suspect it might be a better choice for tasks on which T-CET might fail.
>
>
> | Methods             | CIFAR-10(Acc %) |
> |---------------------|-----------------|
> | Evo(reduced-training)    | 90.32           |
> | Evo(T-CET)          | 96.4            |
> | SG+CCNF+Evo(reduced-training) | 97.3            |
> | SG+CCNF+ZCNAS	| 97.6	|
>
>
> > The method uses a pretrained GPT-Neo-125M model. Hence the actual search cost also implicitly inclues the pre-training cost of this model, which should ideally be added to the search cost computation (table 1,2,3).
>
> We agree there is a hidden cost in needing a pre-trained GPT model - we will include a note about that in the paper for the sake of completeness (note that the authors of GPT-Neo did not disclose the time needed to train their models, so we will rely on how costly it is to train similar ones). However, we are not sure if a straightforward attribution of this cost to our method is completely fair. GPT models are trained to perform NLP tasks and there is no clear relation between those and NAS. This is in contrast to some NAS methods that might require starting from, e.g., an open-sourced pretrained supernet - in this case, the connection to performing NAS is clear and including supernet pretraining cost is justified. The point is: if we consider GPT training cost to be part of the search cost, it gives a false impression of having to pay it each time we search for a new model (which is clearly not the case); even if we consider it a pre-cost, it gives a false impression that it has to be paid each time we want to extend/change our NAS settings (such as including more models in the search space, etc.) - again, this is not the case. As such, we believe it should be classified as “pre-pre-cost”, but there is no clear consensus within the NAS community on how (or even if) such cost should be reported, as far as we know. In particular, many existing methods might have paid analogous cost (i.e., with similar level of indirection) but did think about reporting it. Of course, because of that we are happy to engage in discussion with the reviewer to work out the best way of doing so - please let us know your thoughts about our arguments and the current way in which we include the cost in our paper.

---

> > ### Author Response · Authors · 2023-11-23
> > **Futher Comments**
> >
> > > Study is limited to convolutional spaces. It becomes natural to question the robustness of the proxies to the recent transformer baeed search spaces in NAS, like AutoFormer [1] or HAT [2]?
> >
> > We agree questioning the robustness of proxies is justified, and we also mentioned that in the paper when we talk about the limitations of our work (not only in the context of transformers). As such, extending and testing our method to work with transformers is very sensible. Unfortunately, given the amount of additional experiments, the extra work needed to do so is beyond our current capabilities. For the time being, we will include it in the limitations of our work. However, please note that the core methodology proposed in the paper should be easily compatible with transformers.
> >
> > > Performance on Imagenet is still dominated by methods like Once-For-All which models a simple chain structured space in Table-3 (contrary to the more expressive space here).
> >
> > We have to keep in mind that performance of a network is a function of not only its structure. In particular, existing works have already shown that OFA’s dominating position is heavily dependent on the extensive supernet pretraining (see “OFA scratch” in Tab. 3). For comparison, even if we include the cost of training our final ImageNet model, the total cost of our method is in the range of 400 GPU-hours, compared to 1,200 for OFA’s pretraining.
> >
> > In general, it is known that initialising a subnet from a supernet can bolster final performance. This might seem like a downside of our method, considering our search space is so huge that creating a supernet becomes questionable. However, following on the note at the end of Section 3, we can use any supernet-based NAS on top of our small optimised search space. Due to the time constraints and its excessive training cost, we did not manage to finish running OFA within the rebuttal period - we will include the results in camera ready.
> >
> >
> >
> >
> > > Could the authors ablate the choice of T-CET as a proxy in macro architecture generation across different proxy choices? How robust/sensitive is the search to different choices of (strong) zero-cost proxies?
> >
> > > Could the authors ablate the gains from each of the two phases of hierarchical generative modelling? ie fixing the cell and only performing macro search and vice-versa.
> >
> > The suggested ablations are sensible and we would be happy to include them. However, for the time being, we have prioritised experiments on NB360 and comparison/inclusion of other NAS algorithms (e.g. Evo, more seeds), as we consider them to strengthen the paper in a more fundamental way than ablations. Having said that:
> > 1) we should be perfectly able to add those ablations in time for the camera ready, and
> > 2) we can also share some insights already.
> >
> > For the choice of the ZC proxy, since our search space is a superset of ZenNet space, we strongly expect results to be in line with the performance of each proxy on the ZenNet space, shown in Tab. 2.
> >
> > Regarding micro vs macro: actually the reason why we decided to add the macro stage is the fact that we struggled to obtain competitive results when only generating micro architectures. Allowing cells of different design and different number of channels was crucial to achieve strong performance. How much exactly the gap would be remains to be checked, though (back then we were using our CCNF in a slightly different way, due to the lack of the macro stage). Finally, please note that Tab. 1 already includes some indication of possible results — the models reported there were trained using the same training scheme as those in Tab. 2.
> >
> >
> >
> > > Could the authors study insights derived from NAS method? Which architectural designs tend to be more impactful than the others?
> >
> > We plan to include visualisations of different architectures, but given their size and complexity we struggle a bit in doing so in a legible way.
> > In the meantime, here are some observations about frequent design patterns:
> >       0.   Most of our networks have a “long” skip connection within cells, connecting the input of a cell to the output
> > At the same time, dense connections between operations within cells seem particularly dominant (could be a side effect of simple graphs being very unlikely in the overall space, though)
> > Inverted bottleneck (mobilenet) seems to be the strongest operation and often appears in searched architectures.
> > Similar to Resnext, aggregated residual transformation helps build complex models with fewer parameters
> > Despite the above, the graph topology is relatively similar across different budgets
> > Instead, when we constrain the number of parameters, designs tend to foremost have fewer channels and fewer cells altogether
> > Similarly, when constraining the number of flops, designs tend to include fewer connections between consecutive nodes, leading to more parallel nodes.

---

### Author Response · Authors · 2023-11-23
**Addressing common concerns**

Since many reviewers requested additional experiments of similar nature, we decided to group them all together in a shared response.

> Less-studied tasks/different modalities/generalizability

We decided to run our method on the suit of tasks from NAS-Bench-360 (NB360), which includes diverse realistic tasks (note, we could not finish FSD50k on time due to the dataset size, so it's not included). We compare our results to DASH, a recent method proposed specifically for NB360, and expert networks reported by the benchmark authors.

For running our method, we took #params and #flops of the Wide-ResNet (WRN) baseline as a reference, generated 10 models, selected the best one with T-CET and finally trained it (we were using code provided by DASH to perform training). Results are shown in the table below - all reported values follow the DASH paper (lower is better for all tasks). We can see that although results are less conclusive compared to well-studied tasks, which is expected from a challenging benchmark like NB360, our method remains competitive. Notably, it improves upon DASH, SOTA automated method designed specifically for this benchmark, in 4/8 cases. This is even though we use the same SG+CCNF setup as the one used in other experiments and did not perform any HPO (we were using default parameters for WRN), while DASH performs training and HPO for each of the tasks considered independently. However, the zero-shot nature of our system also means that failure cases, when they happen, are more pronounced. In particular, we can see that on DeepSEA and Satellite our method not only falls short to DASH but is also noticeably worse than the WRN on which we based our conditioning. This is somewhat worrying and certainly indicates some room for improvement. We suspect a sensible way to address this would be to change the final searching algorithm to be training-based so there is a possibility of correcting any serious mistakes. However, running a more time consuming search in place of our cheap selection using T-CET is beyond what we can do in the limited time available for the rebuttal.

| Dataset | Expert | WRN | DASH | Ours |
| - | - | - | - | - |
| Spherical | 67.41 | 85.77 | 71.28 | 71.03 |
| Darcy-flow-5 | 0.008 | 0.073 | 0.0079 | 0.014 |
| Psicov | 3.35 | 3.84 | 3.30 | 3.13 |
| Cosmic | 0.13 | 0.24 | 0.19 | 0.14 |
| Deepsea | 0.30 | 0.40 | 0.28 | 0.50 |
| Ninapro | 8.73 | 6.78 | 6.60 | 6.90 |
| Ecg | 0.28 | 0.43 | 0.32 | 0.32 |
| Satellite | 19.8 | 15.49 | 12.28 | 18.11 |
(note: for ECG we consider our method to improve upon DASH since it achieves the same performance but at a lower cost)


> More seeds

We extended evaluation of our method on ImageNet to report average from 3 searches (3 models, each trained once) and across 3 FLOPs budgets, similar to ZiCo. We show the results in the Table below.

| Budget | Approach | Test acc. (%) |
| - | - | - |
| 450M | ZiCo | 78.1$\pm$0.3 |
| | SG+CCNF | 78.3$\pm$0.2 |
| 600M | ZiCo | 79.4$\pm$0.3 |
| | SG+CCNF | 79.5$\pm$0.14 |
| 1000M | ZiCo | 80.5$\pm$0.2 |
| | SG+CCNF | 80.6$\pm$0.4 |

We can see that although average performance is a bit lower than previously reported, it is still consistently higher than SOTA. At the same time, our method is much faster, considering the cost is amortised across all reported experiments. In general, we hope the reviewers are willing to acknowledge that achieving SOTA on well-established tasks like ImageNet and showing competitive performance on emerging benchmarks like NB360 at the same time, all while enjoying O(1) pre-training cost, is a hard feat to achieve and deserves recognition. To the best of our knowledge, no other method can claim similar performance.

> Code

We will publicly release the code when the paper is accepted. For now, we can share it with the reviewers in a confidential comment.

---

### Author Response · Authors · 2023-11-23
**Code Release**

We would like to inform you that the code for our submission is available for review and reproduction purposes. You can access it at:

https://github.com/visionbasicagent/HGENNAS.

This repository provides the complete implementation details necessary for the reproduction of our results. We hope this will aid in the thorough evaluation of our submission.

---

### Meta-Review · Area_Chair_VXyR · 2023-12-06

**Metareview:**

This work introduces an efficient hierarchical generative modeling for NAS tasks through zero-cost proxies. Overall, the paper is well-written and the reviewers acknowledge the soundness of the method. However, the reviewers also point out that this method largely relies on zero-cost proxies, which can have limited generalization ability. Moreover, the experimental results are not convincing.  The reviewer also suggests to compare with other black box nas methods on the same search space. Considering the mixed opinions and identified weaknesses, I lean towards rejection.

**Justification For Why Not Higher Score:**

Please see the meta-review.

**Justification For Why Not Lower Score:**

N/A

---

### Decision · Program_Chairs · 2024-01-16

Reject